# Development and Application of a Vehicle-Mounted Soil Texture Detector

**DOI:** 10.3390/s20247175

**Published:** 2020-12-15

**Authors:** Chao Meng, Wei Yang, Hong Lan, Xinjian Ren, Minzan Li

**Affiliations:** Key Laboratory of Modern Precision Agriculture System Integration Research, Ministry of Education, China Agricultural University, Beijing 100083, China; mc6663@cau.edu.cn (C.M.); caulanhong@cau.edu.cn (H.L.); sy20193081472@cau.edu.cn (X.R.); limz@cau.edu.cn (M.L.)

**Keywords:** soil texture, soil sensor, electrical conductivity, soil surface image

## Abstract

It is of great significance to obtain soil texture information quickly for the realization of farmland management. Soil with good particle condition can well regulate the needs of plants for water, nutrients, air, and temperature during crop growth, thereby promoting high crop yields. The existing methods of measuring soil texture cannot meet the requirements of time and spatial resolution. For this reason, a vehicle-mounted soil texture detector was designed and developed based on machine vision and soil electrical conductivity devices. The detector does not require pretreatment such as air-drying and screening of the soil, and completely uses the original information of the farmland. The whole process can obtain the soil texture information in real time, omitting the complicated chemical process, and saving manpower and material resources. The vehicle-mounted detector is divided into a mechanical part, a control part, and a display part. The mechanical part provides measurement support for the acquisition of soil texture information; the control part collects and processes signals and images; the measurement results can be intuitively observed and recorded on the display, and can be operated through the mobile phone. The vehicle-mounted detector obtains soil conductivity through 4 disc electrodes, while the vehicle-mounted industrial camera captures the soil surface image, and extracts texture parameters through image processing, takes EC and texture parameters as input, and the embedded SVM model of the instrument was used to perform soil texture prediction. In order to verify the measurement accuracy of the detector, farmland verification experiments were carried out on farmland loam in Tongzhou District and Haidian District of Beijing. The R^2^ of the correlation between the measured value of soil EC and the actual value was 0.75, and the accuracy of soil texture prediction was 84.86%. It shows that the developed vehicle-mounted soil texture detector can meet the requirements for rapid acquisition of farmland texture information.

## 1. Introduction

Soil texture is one of the important physical properties of soil. It represents the percentage combination of soil particles of different diameters in soil weight. Soil texture has an important impact on soil fertility, quality, and sustainable use of soil. Soil with good texture can well regulate the needs of plants for water, nutrients, air, and temperature during crop growth, thereby promoting high crop yields [1]. The soil texture is mainly divided into three categories: Sandy soil, loam, amd clay. Among them, loam has the advantages of sand and clay which is an ideal soil texture type, suitable for planting crops, especially peanuts, tobacco leaves, and vegetables.

The standard method for obtaining soil texture is the straw method. Although this method has high accuracy, it takes a lot of time, and requires heating and using H_2_O_2_ and Na_3_PO_4_. The whole process requires manual operation to ensure accuracy and safety. The measurement accuracy depends on experimental conditions and the proficiency level of the operator [2]. For the acquisition of soil texture information, some new methods have been developed in recent years, such as: Gamma ray method, sieve analysis method, laser diffraction method, and scanning electron microscopy method. These methods require specific expensive instruments, and most of them also need soil pretreatment. And the preprocessing cannot achieve real real-time fast measurement [3,4,5]. In addition, there are many novel researches for the acquisition of soil texture information. For example, Vos [6] pointed out that manual identification based on experience can replace laboratory analysis. This method only relies on hand feeling and completely depends on the experience of the operator and is not universal. Hobley [7] used vis-NIR technology to estimate texture, and B. Jović [8] used Diffuse Reflaxions Infrared Fourier Transformations Spectroscopy (DRIFT) to obtain spectra and determined the spectral characteristics of five typical soil types in a certain area. However, both of these methods require specific expensive equipment. Wu [9] used remote sensing technology to identify soil types, and the results showed that the model with NDVI plus topography and stratum performed best with overall accuracy, kappa statistic, and area under the curve of 0.975, 0.918, and 0.907. Many scholars have used multi-spectral and hyperspectral satellite data to classify regions with different soil textures [10,11,12,13]. Wu and Castaldi used BJ-1 [14] and Advanced Land Imager (ALI) satellite imager [15] to quantitatively estimate the content of sand and clay in the soil, and they performed well. However, the current research has time lag in the acquisition of soil texture, and the spatial resolution is also difficult to meet the demand, and most of them require expensive experimental equipment. In summary, there is an urgent need for a low-cost, real-time, and high-accuracy soil texture information acquisition method and detector.

Scholars at home and abroad have studied the use of other soil parameters to predict soil texture. Some studies use soil electrical conductivity (EC) to characterize soil texture, EC has a wide range of applications in soil research. It can be used to study the dissociation and exchange performance of adsorbed ions in the soil, and reveal the strength of interaction between various ions and soil colloids. Many studies have shown that EC can reflect soil texture to a certain extent [16]. Heil [17] uses soil EC to characterize soil texture variability at highly variable locations. This study describes the soil texture by combining point-by-point EC measurements with digital terrain model, cultivation parameters, and the thickness of the Quaternary sediments. R^2^ values ranged between 0.67 and 0.76. This shows to a certain extent that the combination of EC and other soil characteristics has a good correlation with soil texture. There are also studies using soil surface images to predict soil texture or classify soil. Jia [18] used hyperspectral image technology to establish a classification model by combining effective wavelength and texture feature data to predict the soil types of red soil, paddy soil, and seashore saline soil, with a correct rate of 90%. Although this method is only for soil type and color, it provides an idea for using images to classify soil texture. Morais [19] used a multivariate image analysis method to predict soil texture with a very high success rate when the soil composition is low in powder. However, even if this method is twice as long as the standard pipette analysis method, it still takes 50 h and cannot quickly measure soil texture. Predecessors in the same laboratory have shown that using image methods to obtain soil characteristics and predict soil roughness can provide a reference for analyzing soil texture. Therefore, we can try to use soil EC and soil surface images for data fusion to quickly predict soil texture.

In fact, it can be further explained that the fusion of these two kinds of data can be used as the basis for determining soil texture: The EC of the soil can reflect the size of the parameters such as soil texture and porosity to varying degrees. Similarly, the texture features extracted from the soil surface image have a strong relationship with the soil surface roughness [20,21,22,23,24]. Li [25] put forward a method to interpret soil roughness by taking field photos, which can obtain soil roughness through soil surface image. García [26] used image shading to obtain soil surface roughness in the laboratory and on-site. The soil surface roughness is a parameter that characterizes the micro-topography of the soil. The small spacing of soil surface particles and the unevenness of small peaks and valleys are related to the size of the soil surface particles. The different combinations of soil particle size are soil texture. These two kinds of data are closely related to texture. Scholars at home and abroad have conducted preliminary studies on the methods of using soil EC and soil surface image texture features to predict soil texture. Therefore, this study proposes to use soil EC and soil surface image texture features for data fusion to predict soil texture.

This research aims to develop a vehicle-mounted soil texture information acquisition instrument that uses the original information of farmland, and to verify the instrument through field experiments. The methods are as follows: Using disc electrode to obtain soil EC, using industrial camera to obtain soil surface image and extracting texture parameters, and establishing model and prediction based on support vector machine.

## 2. Materials and Methods

### 2.1. Soil Texture Prediction Method

Because soil texture is difficult to measure directly, a method for predicting soil texture using parameters that are highly correlated with soil texture is proposed. The above analyzed the feasibility of indirect prediction of soil texture, and finally chose the method of combining soil EC and soil surface image to predict soil texture. First, using soil EC data and the texture feature information of the soil surface image and soil texture to build a model, and using soil from different regions as input to make the model more versatile. In farmland experiments, through the designed soil EC sensor and industrial camera, the required input parameters: Soil EC and texture features extracted from the soil surface image are obtained respectively, and then the prediction model is called to obtain the soil texture information of the target farmland. A brief flow chart of soil texture prediction is shown in Figure 1.

### 2.2. Soil EC Measurement Principle

The current-voltage four-terminal method is the most classic method for measuring soil EC. The principle structure diagram is shown in Figure 2. A constant current source provides a constant current between J and K, and the voltage between M and N is measured by a voltmeter. The voltage drop at both ends is calculated, and the soil EC is calculated by this voltage drop [27].

When a conductor has a regular shape, its EC value is a function of cross-sectional area and length. However, soil is composed of various granular minerals, organic matter, moisture, air, microorganisms, etc., which is a very irregular and complex substance. It is a porous dispersion medium composed of countless soil particles arranged in layers, so the cross-sectional area and length cannot be measured, so there is a soil EC calculation formula for the structure of the soil [28]:(1)$$\sigma_{\mathrm{MN}}=\frac{\frac{1}{d_{\mathrm{JM}}}-\frac{1}{d_{\mathrm{JN}}}-\left(\frac{1}{d_{\mathrm{KM}}}-\frac{1}{d_{\mathrm{KN}}}\right)}{2\pi }\frac{I}{V_{\mathrm{MN}}}=K\frac{I}{V_{\mathrm{MN}}},$$
$\sigma_{\mathrm{MN}}$—Calculated soil EC (μS/cm)$d_{\mathrm{JM}}$, $d_{\mathrm{JN}}$, $d_{\mathrm{KM}}$, $d_{\mathrm{KN}}$—The distance between the probes (cm)I—Constant current source current (A)$V_{\mathrm{MN}}$—Voltage between M and N probes (V)

When the output of the steady amplitude Alternating Current source is constant, the soil EC is in inverse proportion to the voltage drop at the voltage terminal.

In particular, a constant current source using a direct current excitation method is prone to polarization, so alternating current is usually used. Moreover, the constant current source cannot always be kept constant in the actual experiment. It will fluctuate due to the change of the load. Since the ground is irregular and its load changes all the time, the four-terminal method is usually improved as shown in Figure 3: An ammeter is connected in series with the constant current source circuit to obtain the actual value of the constant current source to ensure the accuracy of the current-voltage four-terminal method [29].

### 2.3. Principles of Soil Surface Image Analysis

The method of extracting soil surface texture features is gray level co-occurrence matrix method (GLCM). GLCM [30,31,32,33] is a classic second-order statistical algorithm. In 1973, Haralick proposed the use of a GLCM to describe texture features. This is because the texture is formed by the grey distribution repeatedly and alternately in the spatial position, so there must be a certain distance between two pixels in the image space. A certain grey level relationship is called the spatial correlation characteristics of grey levels in an image. The texture is described by studying the spatial correlation of grey levels. This is the ideological basis of the GLCM. It is composed of the joint probability density of gray levels. It can reflect the comprehensive information about the direction, adjacent interval, and change range of the image gray level. It is the basis for analyzing the local patterns of images and their arrangement rules. Based on the matrix, a variety of statistics can be calculated: Energy, entropy, contrast, uniformity, correlation, variance, sum average, etc. The most used parameters are as follows [34]:

Correlation: Correlation refers to the degree of correlation between related pixels and their neighboring pixels, which reflects the local gray-scale correlation in the image.
(2)$$\mathrm{Cor}=\sum_{i}\sum_{j}\frac{P\left(i,j\right)}{1+{\left(i-j\right)}^{2}},$$
P—Image(i,j)—Pixel coordinates

Energy: Energy measures the uniformity of the texture of an image and represents the repetitive information of pixel pairs.
(3)$$\mathrm{Eng}=\sum_{i}\sum_{j}P{\left(i,j\right)}^{2},$$
P—Image(i,j)—Pixel coordinates

Entropy: The meaning of entropy in physics is the degree of regularity of an object. The more orderly, the smaller the entropy, and the more disorderly, the greater the entropy. It represents the randomness and complexity of texture feature distribution.
(4)$$\mathrm{Ent}=\;\sum_{i}\sum_{j}P\left(i,j\right)\log P\left(i,j\right),$$
P—Image(i,j)—Pixel coordinates

In actual use, the required texture parameters are selected for extraction according to the situation, but in most cases, the above parameters are directly used. After the research of predecessors in the laboratory, energy, homogeneity, entropy, energy, and moment of inertia have a high correlation with soil roughness, soil bulk density, root mean square height, and correlation length. By comparing the correlation between the root–mean–square height and the correlation length that characterize the soil surface roughness and the 12 texture feature parameters, the 4 texture features used in this article have the highest R^2^. The R^2^ of the root–mean–square height and the correlation length with the four texture parameters are as follows: The R^2^ of energy is 0.66 and 0.52; the R^2^ of entropy is 0.72 and 0.63; the R^2^ of momentum of inertia is 0.71 and 0.68; the R^2^ of correlation is 0.7 and 0.62. Therefore, this study uses four image parameters: Energy, entropy, moment of inertia, and correlation to predict soil texture [35].

### 2.4. Detector Design

Based on the measurement principle and method of soil EC and soil surface image, the detector is developed. The detector structure is shown in Figure 4. The whole structure is connected to the tractor by a rear-mounted three-point suspension. Two depth-limiting wheels on both sides can work together with the tractor hydraulic lifter to adjust the depth of soil. The hardware detector is mainly composed of three parts: 1. Disc electrode for measuring EC; 2. Industrial camera to obtain soil surface image; 3. Detector circuit (including constant current source generating circuit, data acquisition card, and GPS). The physical picture of the detector is shown in Figure 5. Two equipment boxes are designed on the beam to be placed symmetrically on both sides, one of which is placed with 12 V power supply lithium batteries and circuit modules, and the other is placed with equipment related to industrial cameras. The equipment is fixed in the equipment box with screws and custom-sized concave iron sheets to alleviate the impact of vibration, effectively prevent soil dust and rain from damage to the equipment, and increase the counterweight to make the sensor electrode fully contact the soil.

The soil EC part (1, 2, 3, 5 in Figure 4) has four disc electrodes, the diameter of the disc electrodes is 20 cm, and they are in close contact with the soil when they enter the soil. Since the disk electrode can roll easily, compared with the non-disk electrode, it can reduce the interference caused by soil resistance, improve the stability of the signal, and can measure soil EC information more accurately. When measuring the soil, adjust the depth limit wheel to ensure that 2/5 of the disk electrode is in the soil. The entire detector follows the tractor. The four disk electrodes are inserted into the soil to roll, and two external electrodes are used as constant current source output Electrodes, the two internal electrodes send the measured electrical signals to the system circuit through the brushes on both sides of the electrodes, and display them on the industrial flat panel after processing and calculation.

An industrial camera was used to get the image of soil surface (1, 7 in Figure 4). The industrial camera is detachable, and the position can be adjusted according to the actual farmland conditions. It has 10 million pixels (3664 × 2748) and a frame rate of 8. The aforementioned industrial camera is used to take an image of the soil surface and perform image processing to extract GLCM texture features in the whole system. According to the height and depth limit of wheel and the tractor, the image area is generally 20 cm × 20 cm. The centerline position of the four side-by-side disk electrodes is regarded as the EC measurement position, and the industrial camera is installed above the centerline.

These two parameters are input into the embedded model to predict the soil texture of the target plot.

The collected soil information can be browsed through industrial tablets or viewed through mobile phones. The mobile phone display interface is shown in Figure 6, which can view, manage, and analyze soil data.

The principle diagram of the detector is shown in Figure 7. The most important part of the whole detector is the data collection of two kinds of sensing devices: Disc type EC electrode and industrial camera. Four disc electrodes were used for measuring soil EC. Industrial camera was used for acquiring soil surface images. Among them, the industrial camera and the industrial tablet computer are directly connected through a USB cable. In order to achieve real-time and rapid data measurement, a high-speed data acquisition card is selected to ensure the flow and accuracy of the data, and a GPS receiver is also installed and used to record position information

### 2.5. Soil Samples and Experimental Preparation

The geographical coordinates of Beijing are 115.7~117.4° East longitude, 39.4~41.6° N latitude, with an area of 16,410 km^2^, an annual average temperature of 11.5 °C, and a precipitation of about 540.7 mm. The soil is mostly gray loess and brown gray soil. The soil is fertile and moderate in texture. The main soil texture is loam. The experiment site was in two districts of Beijing: Tongzhou and Haidian, and the soil texture is loam.

The distribution of experimental plots, paths and sampling plots is shown in Figure 8. The two farmlands are both 2500 m^3^ corn fields, and the sampling plot is 2.5 m × 2.5 m. Limited by the height of the camera, the shooting area is smaller than the area of the sampling cell and the conductivity measurement area, so shooting at the center of the sampling cell approximately represents the entire cell. The experiment content includes data collection and soil sample collection. A total of 185 soil samples were collected on the experimental path, including 100 in Tongzhou and 85 in Haidian, which were matched with measured values through GPS data. Each soil sample is 1 kg and the sampling depth is 10 cm. Each soil sample is divided equally by the 4-point method to measure the standard value of EC and soil texture.

Each soil sample is used to measure EC. The laboratory measurement method [36] is: Place the soil sample in the tray to air dry and pass through a 1 mm sample sieve. Weigh 10 g of the soil sample in a shaker bottle and add 50 mL of deionized water at 20 °C. Cover the bottle cap and place it on a reciprocating horizontal constant temperature oscillator to oscillate for 30 min. After oscillating, let it stand at 20 °C for 24 h. Take an appropriate amount of clear solution and measure it with a EC instrument. Record the data as the standard EC value of the soil sample.

Three soil texture types were measured, namely sandy loam (63 samples), light loam (97 samples) and medium loam (25 samples). The measurement method is: Using a laser particle size analyzer (NKT5200-H, Shandong Nikeite Analytical Instrument Co., Ltd., China) to perform wet measurement on the air-dried and sieved soil sample. After obtaining the analysis report, the soil samples are divided into sandy loam soil, light loam soil, and medium loam soil according to the Kaczynski soil classification standar [37] in Table 1.

In fact, the three texture types are relatively similar in particle size composition ratio. When the particle size of less than 0.01 mm is in the range of 10% to 20%, it is sandy loam, 20% to 30% is light loam, and 30% to 40% is medium loam. Therefore, it is difficult to distinguish the three kinds of soil texture manually through experience, and it is necessary to use a detector to measure.

## 3. Results

### 3.1. Electrical Conductivity and Texture Features

The descriptive statistics of the samples of sandy loam, light loam, and medium loam are shown in Table 2, including GLCM texture parameters and EC. The average EC values of sandy loam, light loam and medium loam were 271 μs/cm, 225 μs/cm, and 218 μs/cm respectively. As the percentage of soil physical clay particles (<0.01 mm) increased, EC showed a decreasing trend. Except for the EC data, the standard deviations are small, indicating that the data except for the EC are closer to the average and the data is relatively stable. Among them, the Correlation of sandy loam soil and light loam soil and the energy of sandy loam soil have larger absolute values of kurtosis, which are quite different from the normal distribution, except that they are basically in line with the normal distribution. The absolute value of skewness is small, indicating that the left and right sides of the dispersion are relatively even.

The average value of GLCM texture features extracted from 185 sample points is shown in Figure 9. The samples are divided into 3 texture types for comparison. The four pictures are respectively energy, entropy, moment of inertia, and correlation of the four texture feature values. The abscissas in the figure are 0°, 45°, 90°, and 135°, respectively, to extract the four feature value directions. It can be seen that the 4 texture values of the 3 texture types have obvious differences, and the change trend is regular, indicating that these 4 texture parameters have obvious help for texture classification. For entropy and moment of inertia, sandy loam, light loam, and medium loam show a decreasing trend. On the contrary, for energy and correlation, sandy loam, light loam, and medium loam show an increasing trend. This shows that the sandy loam soil is rougher, which is consistent with the actual situation.

### 3.2. Analysis of Soil EC Measurement Results

EC measurement is a key part of the vehicle-mounted soil texture detector. In order to verify the accuracy of EC measured by the detection instrument, a linear regression model of soil EC was established by using the 185 EC values actually measured by the detector in the farmland and the standard EC values measured by the laboratory extraction method. The result is shown in Figure 10.

It can be seen that the measured values are evenly distributed on both sides of the regression line, and the correlation analysis result R^2^ is 0.75. The results show that the soil EC data obtained by the vehicle-mounted soil texture detector has high accuracy.

### 3.3. Analysis of Soil Texture Measurement Results

The prediction model embedded in the vehicle-mounted soil texture detector is a support vector machine (SVM) model. SVM has been proved to have outstanding advantages in the case of small sample sizes, which can avoid the problem of neural network structure selection and local minimum points, and has excellent learning performance and good robustness [38]. The kernel function used by the built-in model of the detector is the RBF function, and the optimization results of the model parameters are: C is 50.61 and g is 2.93.

In order to verify that the combination of the two parameters of soil EC and GLCM texture features extracted from soil surface images is the optimal input, the following experiments were carried out: Soil texture is predicted only by soil EC; soil texture is predicted by using GLCM texture feature extracted from soil surface image; soil is predicted by combining soil EC with GLCM texture feature extracted from soil surface image Texture. The results are shown in Table 3.

The GLCM texture parameters extracted from soil surface images and soil EC are input to the built-in SVM model of the detector. The soil texture measurement value predicted by the vehicle-mounted soil texture detector is analyzed with the standard value measured by the laser particle size analyzer. The correct rate of the tester’s measurement was verified, and the result is shown in Figure 11. The correct rate is the ratio of the number of correct samples to the total number of samples.

The *x*-axis in Figure 11 is the serial number of the soil sample. In order to display the measurement results more directly in the figure, the samples were not sorted according to the number of the samples collected during the experiment, but 185 samples were re-sorted according to the type of soil texture. The first 63 samples are sandy loam soil, the last 25 samples are medium loam soil, and the remaining 97 samples are light loam soil.

Among them, the prediction accuracy rate of only using soil EC to predict soil texture is 56.21% (104/185); only using GLCM texture features extracted from soil surface images to predict soil texture is 78.38% (145/185); using soil EC and soil surface The combination of GLCM texture features extracted from the image has the highest accuracy in predicting soil texture, with a total accuracy of 84.86% (157/185). Among them, the prediction accuracy rate of sandy loam and light loam is above 87%. The correct rate of medium loam soil was 64%, and 9 samples were wrongly judged as light loam soil. The reasons may be: In the Kaczynski soil texture classification standard, the content of physical clay particles (<0.01 mm) is light loam when the content is 20~30%, and the content is medium loam when the content is 30~40%. The content of physical clay (<0.01 mm) is about 30.5%, which is on the dividing line of the two textures, and the difference is small, so the judgment is wrong.

The results show that the vehicle-mounted soil texture detector can use the original farmland information to predict the soil texture with high accuracy, and the combination of soil EC and soil surface image is the optimal input.

## 4. Discussion

First of all, the detector is designed and developed on the principle of soil EC and surface texture characteristics. The hardware part is stable and meets the needs through observation and inspection during the experiment. The results of field experiments show that the combination of EC and GLCM texture is the best input, and the detector has high accuracy in soil texture measurement.

In the current research, the first-order statistical features (such as skewness and kurtosis) are only related to the attributes of a single pixel, and cannot reflect the spatial relationship of pixels in the image, so the gray-scale method is used to propose the second-order statistical texture feature [39]. Chen [40] used GLCM texture extracted from simulated tumor images to distinguish true progression and false progression of glioblastoma treated with radiotherapy and temozolomide, with an accuracy of 86.4%, and found that correlation was related to abnormal inhomogeneity, and texture characteristics of energy and entropy were related to local homogeneity, which was consistent with the experimental results of this study. Among them, GLCM texture is widely used in image-based classification research. Ou [41] used GLCM to conduct in vivo skin capacitance imaging analysis. He found that because the texture on three different skin positions became denser with the increase of the number of tape peelings, the entropy value was at the three different skin positions. It shows an upward trend, which is consistent with the results of this study. As the soil particle composition becomes finer, the entropy value gradually decreases. Cho [42] studied a linear regression model for estimating clay texture content with EC as an independent variable. This model provides a good degree of fit on the surface, but the degree of fit decreases with depth, from 0.84 to 0.23. The study shows that EC It contributes to the prediction of soil texture, but it cannot be explained by simple rules. This is also consistent with the viewpoint expressed by Kelleners [43]. Mahmood [44] used EM38 to predict soil texture. In the farmland with 37% clay content, R^2^ measured by EM38 and clay, loam and sand are 0.28, 0.34, and 0.39 respectively. In the farmland with very low clay content (5%), the R^2^ measured by EM38 and clay, loam and sand were 0.49, 0.09, and 0.05, while in the farmland with an average clay content of 19%, the R^2^ measured by EM38 and clay, loam and sand were 0.2, 0.71 and 0.72, respectively. It can be seen that EC and texture have a certain correlation, but how they affect each other is unknown. In fact, all kinds of soil physical parameters are interrelated and restricted each other, so it is impossible to predict soil texture comprehensively and accurately by single parameter, when studying EC and soil texture, it is necessary to assume uniform fertility [45], but this is not noticed in most studies, and the fertility of collected soil samples is indeed difficult to actively select and control, so the low correlation between EC and texture and the difficulty in summarizing the law can be attributed to this.

The detector is vehicle mounted and suitable for large-scale farmland or farmland with different crops in each small area. In particular, most of China’s farmland is small area, but the aggregation is dense, and the single farmland area is small, but the aggregation scale becomes larger. The farmland in a single small area cultivates different crops, so it is more suitable to use vehicle mounted detector to measure the soil texture, which can obtain the texture type of the region and give the most suitable suggestions for planting a certain crop.

In addition, the detector also has limitations. Due to the unevenness and heterogeneity of the soil, the image may be affected by weather conditions (such as rain, fog, snow, drought, and cloudy). For example, it may have a negative impact on the camera’s focus, or white balance and exposure issues under strong light conditions. Therefore, the experiments in this study have manually avoided the harsh environment and selected suitable weather and experimental farmland. We plan to expand research in the future to enhance the ability of the detector to adapt to different environments.

Finally, the detector provides a reference for measuring soil physical parameters using only the original information of farmland. We also plan to add a deep loose plow to the detector in the future to estimate soil bulk density and total porosity. For example, first predict the texture of the target plot, and then predict the soil bulk density and total porosity based on the texture type to increase accuracy.

## 5. Conclusions

In this study, a vehicle-mounted soil texture detector was designed and developed based on soil EC and soil surface image. Four disc electrodes are used to obtain the EC by the current-voltage four-terminal method, and the industrial camera is used to capture the soil surface image and extract the GLCM texture feature. The EC and GLCM are input into the embedded SVM model of the detector to obtain the soil texture information of the target plot.

Based on the feasibility analysis of the texture measurement principle, the soil EC is combined with the machine vision device, and the SVM model is used as the embedded model to obtain and analyze the in-situ texture information of farmland soil in real time. Compare the results obtained by this method with the standard method data. The results show that this method of obtaining in-situ texture information of farmland soil does not require chemical reagents, long test time, and artificial energy, and it is a new method to quickly obtain soil texture in real time.The correlation analysis between the farmland measurement results of the in-situ vehicle-mounted soil texture detector and the results obtained by the laboratory standard method is carried out, and the correlation analysis result R^2^ of the soil EC measurement is 0.75. The accuracy rate of soil texture prediction data obtained by combining EC and GLCM using the embedded model reached 84.86%. The results show that the vehicle-mounted soil texture detector combined with EC and GLCM can predict the soil texture of the target plot based on the original information of the farmland, and has high accuracy.

## Figures and Tables

**Figure 1 sensors-20-07175-f001:**
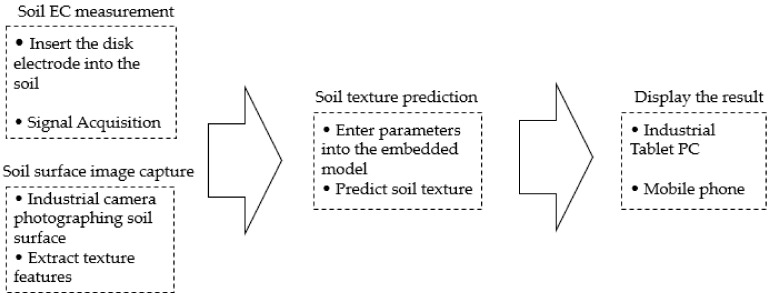
Flow chart of soil texture prediction.

**Figure 2 sensors-20-07175-f002:**
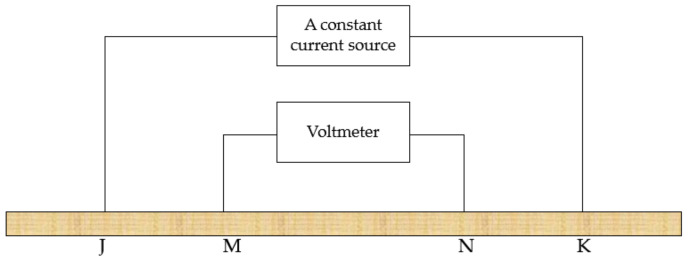
Schematic diagram of current-voltage four-terminal method.

**Figure 3 sensors-20-07175-f003:**
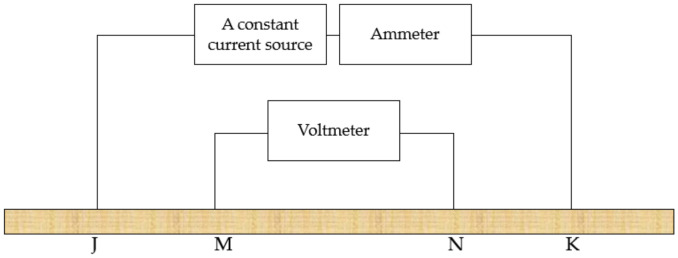
Improved current-voltage four-terminal method schematic diagram.

**Figure 4 sensors-20-07175-f004:**
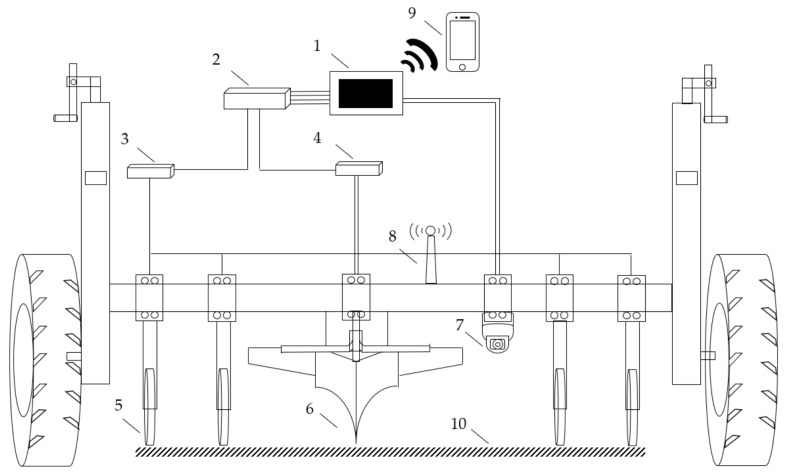
Detector structure diagram. 1 Industrial tablet computer; 2 Data acquisition card; 3 Circuit 1; 4 Circuit 2; 5 Disc electrode; 6 Deep loose plough; 7 Industrial camera; 8 GPS locator; 9 Mobile phone; 10 Centerline.

**Figure 5 sensors-20-07175-f005:**
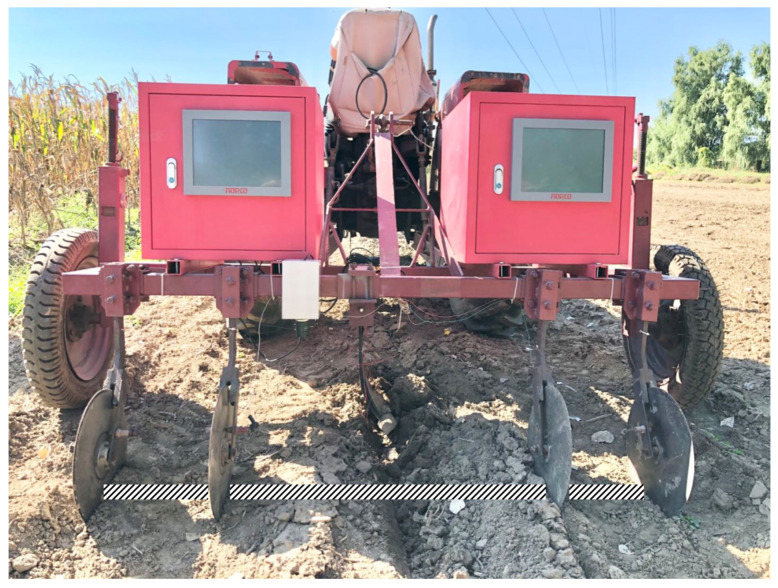
Physical image of detector.

**Figure 6 sensors-20-07175-f006:**
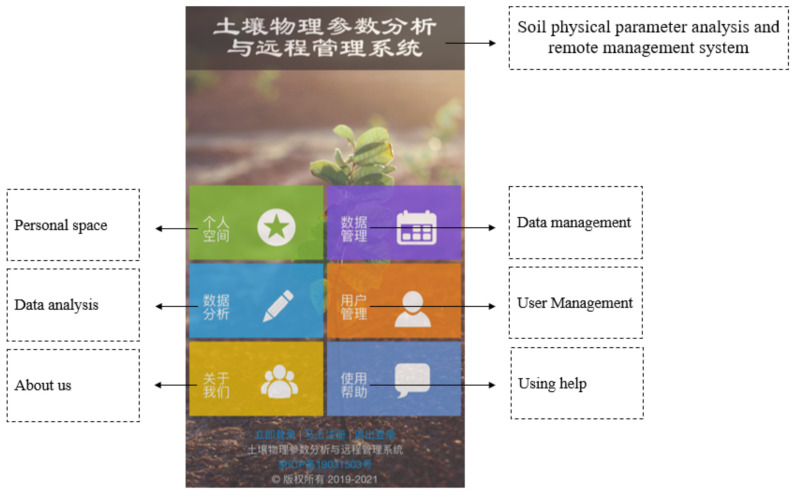
Mobile phone interface.

**Figure 7 sensors-20-07175-f007:**
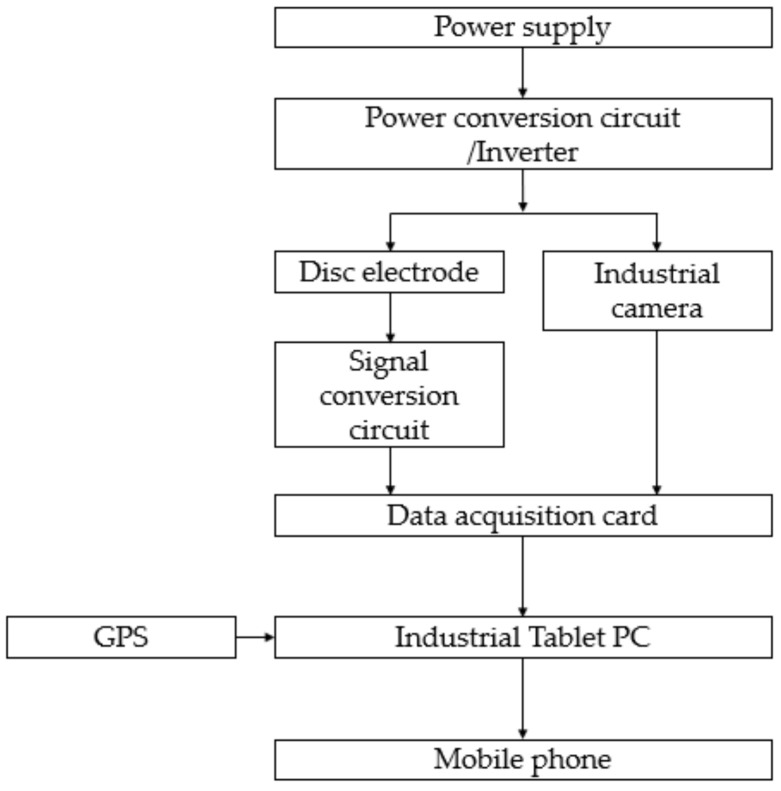
Schematic diagram of detector.

**Figure 8 sensors-20-07175-f008:**
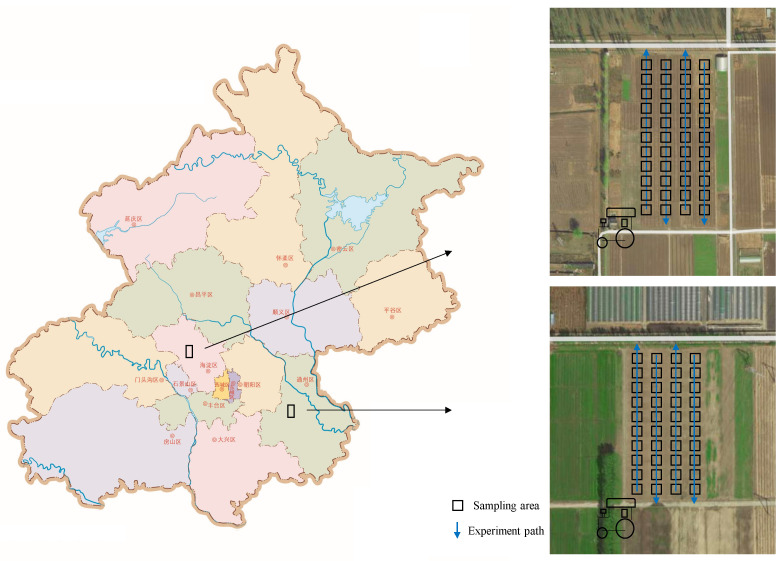
Experimental plots, paths, sampling plots.

**Figure 9 sensors-20-07175-f009:**
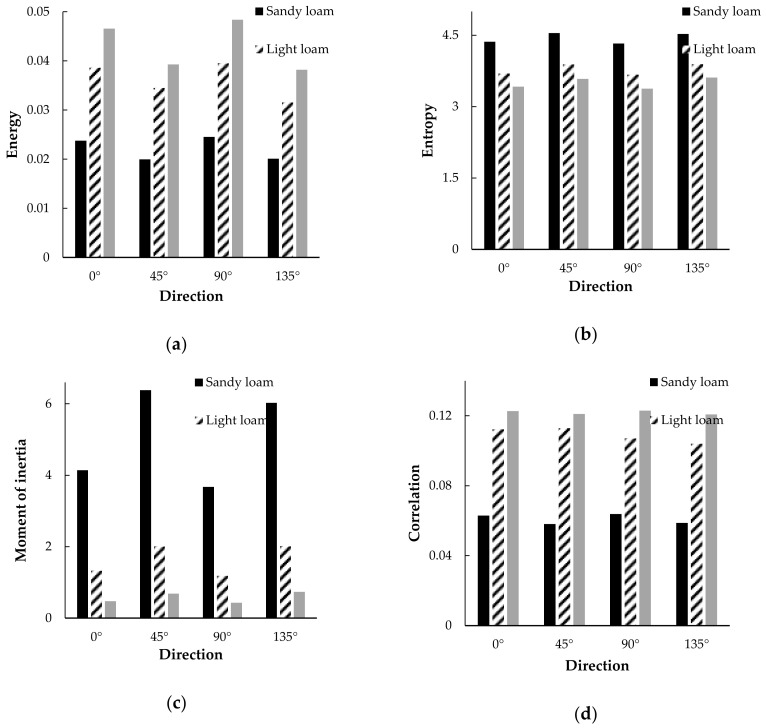
GLCM texture feature average map. (**a**) Average of Energy; (**b**) Average of Entropy; (**c**) Average of Moment of inertia; (**d**) Average of Correlation.

**Figure 10 sensors-20-07175-f010:**
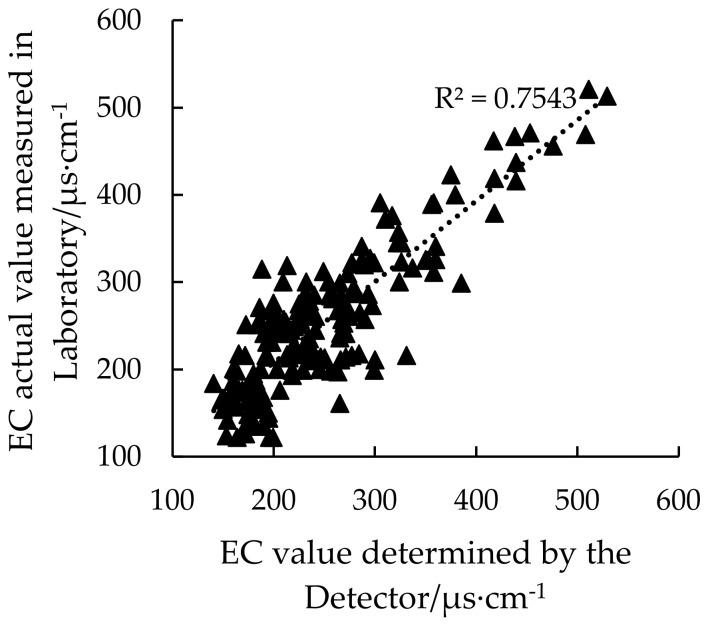
Soil EC measurement results.

**Figure 11 sensors-20-07175-f011:**
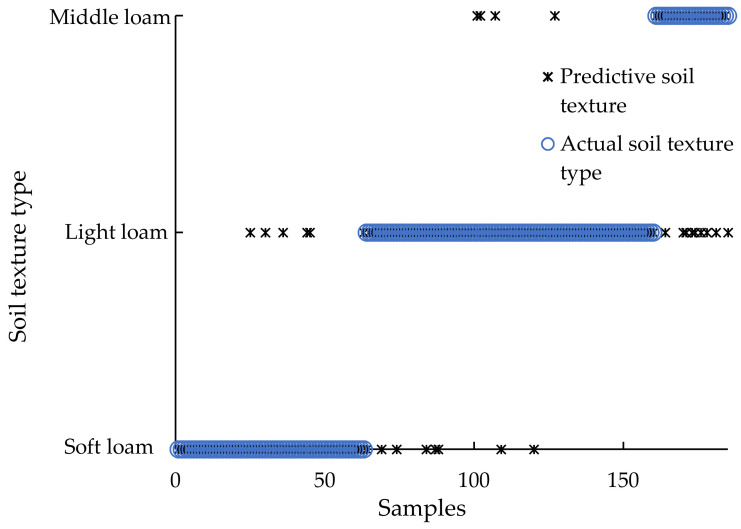
Soil texture measurement results.

**Table 1 sensors-20-07175-t001:** Kaczynski’s standard for classification of soil texture.

Soil Texture		Physical Clay (<0.01 mm) Content			Physical Clay (>0.01 mm) Content		
		Podzol	Grassland Soil, Red and Yellow Soil	Columnar Alkaline Soil, Strong Alkaline Soil	Podzol	Grassland Soil, Red and Yellow Soil	Columnar Alkaline Soil, Strong Alkaline Soil
Sand	Loose sand	0–5	0–5	0–5	100–95	100–95	100–90
	Tight sand	5–10	5–10	5–10	95–90	95–90	95–90
Loam	Sandy loam	10–20	10–20	10–15	90–80	90–80	90–85
	Light loam	20–30	20–30	15–20	80–70	80–70	85–80
	Middle loam	30–40	30–45	20–30	70–60	70–55	80–70
	Heavy loam	40–50	45–60	30–40	60–50	55–40	70–60
Clay	Light clay	50–65	60–75	40–50	50–30	40–25	60–50
	Medium clay	65–80	75–85	50–65	35–20	25–15	50–35
	Heavy clay	>80	>85	>65	<20	<15	<35

**Table 2 sensors-20-07175-t002:** Sample descriptive statistics table. GLCM—gray level co-occurrence matrix method; EC—electrical conductivity.

Texture	GLCM and EC	Mean	Standard Error	Median	Standard Deviation	Kurtosis	Skewness
Sandy loam	Energy	0.0221	0.0021	0.0190	0.0159	5.7348	2.3370
	Entropy	4.4408	0.0602	4.5656	0.4662	2.6134	−1.6601
	M of I	5.0549	0.3172	5.8599	2.4570	−0.8623	−0.4730
	Correlation	0.0609	0.0039	0.0522	0.0301	28.9858	4.9598
	EC	271.665	8.9177	257.000	69.0765	0.3877	0.9216
Light loam	Energy	0.0360	0.0021	0.0326	0.0202	17.2707	2.7564
	Entropy	3.7861	0.0490	3.6955	0.4800	−0.0321	0.6633
	M of I	1.6319	0.2023	0.7209	1.9823	2.8862	2.0293
	Correlation	0.1091	0.0067	0.0966	0.0652	20.0478	3.0386
	EC	225.0635	6.3721	201.500	62.4339	1.5693	1.2774
Middle loam	Energy	0.0431	0.0029	0.0376	0.0145	−0.1002	0.7838
	Entropy	3.4965	0.0556	3.5691	0.2779	−1.1965	−0.2736
	M of I	0.5814	0.0411	0.5107	0.2053	−1.4442	0.3729
	Correlation	0.1219	0.0096	0.1046	0.0479	1.5727	1.3814
	EC	218.784	11.676	199.300	58.3801	1.3856	1.2557

**Table 3 sensors-20-07175-t003:** Soil texture measurement result table.

Input			Predicted Soil Texture			
			Sandy Loam	Light Loam	Middle Loam	Correct Rate
	Actual soil texture	Sandy loam	32	31	0	50.79%
EC		Light loam	25	72	0	74.23%
		Middle loam	4	17	4	16%
	Total correct rate	-	-	-	-	56.21%
	Actual soil texture	Sandy loam	52	10	1	82.59%
GLCM		Light loam	11	85	1	87.63%
		Middle loam	0	15	10	40%
	Total correct rate	-	-	-	-	78.38%
	Actual soil texture	Sandy loam	56	7	0	88.89%
EC and GLCM		Light loam	8	85	4	87.63%
		Middle loam	0	9	16	64%
	Total correct rate	-	-	-	-	84.86%

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
