# Peer review of "Development and Application of a Vehicle-Mounted Soil Texture Detector"

_sensors, 2020, doi:10.3390/s20247175_

Round 1
Reviewer 1 Report
Please find enclosed a pdf file consisting of the main text of the paper and my remarks and comments.
Thank you.

Author Response
Response to comments from reviewers
Thank you very much for your helpful comments and advice.
sensors-997167
Title: Development and application of a vehicle-mounted soil texture detector

Reviewer 2 Report
- need to point out how to do the ground-truth correlation of EC and soil texture.
- Table 3, it seems the R2 (Coefficient of determination)of EC and soil texture is not credible in the hypothesis.
- also how to mapping the soil surface images to soil texture needs to be presented. it seems the image is the main contribution to detecting texture type.
- L322, The GLCM texture parameters extracted from soil EC and soil surface images are input to the built-in SVM model of the detector. do you mean: The GLCM texture parameters extracted from soil surface images and soil EC are input to the built-in SVM model of the detector.
- L329, do you mean: sorted and stored
- repeated Fig 10.
- L388, Due to the unevenness and unevenness of the soil
Author Response

(The authors gave the same response as above.)

Round 2
Reviewer 1 Report
I accept your corrections. I recommend the text for publication after polishing of English language.
Reviewer 2 Report
The author has stated most of the doubts, and therefore there is no need for further modification.